# Natural Products and Their Derivatives against Human Herpesvirus Infection

**DOI:** 10.3390/molecules26206290

**Published:** 2021-10-18

**Authors:** Chattarin Ruchawapol, Man Yuan, Si-Min Wang, Wen-Wei Fu, Hong-Xi Xu

**Affiliations:** 1School of Pharmacy, Shanghai University of Traditional Chinese Medicine, Cai Lun Lu 1200, Shanghai 201203, China; chattarin.ruchawapol@hotmail.com (C.R.); peggyyuan1990@163.com (M.Y.); ll19821267503@163.com (S.-M.W.); 2Engineering Research Centre of Shanghai Colleges for TCM New Drug Discovery, Shanghai 201203, China

**Keywords:** herpesviruses, alkaloids, flavonoids, terpenoids, polyphenols, anthraquinones, anthracyclines, anti-herpes virus activity

## Abstract

Herpesviruses establish long-term latent infection for the life of the host and are known to cause numerous diseases. The prevalence of viral infection is significantly increased and causes a worldwide challenge in terms of health issues due to drug resistance. Prolonged treatment with conventional antiviral drugs is more likely to develop drug-resistant strains due to mutations of thymidine nucleoside kinase or DNA polymerase. Hence, the development of alternative treatments is clearly required. Natural products and their derivatives have played a significant role in treating herpesvirus infection rather than nucleoside analogs in drug-resistant strains with minimal undesirable effects and different mechanisms of action. Numerous plants, animals, fungi, and bacteria-derived compounds have been proved to be efficient and safe for treating human herpesvirus infection. This review covers the natural antiherpetic agents with the chemical structural class of alkaloids, flavonoids, terpenoids, polyphenols, anthraquinones, anthracyclines, and miscellaneous compounds, and their antiviral mechanisms have been summarized. This review would be helpful to get a better grasp of anti-herpesvirus activity of natural products and their derivatives, and to evaluate the feasibility of natural compounds as an alternative therapy against herpesvirus infections in humans.

## 1. Introduction

Herpesviruses (Family: Herpesviridae) are enveloped viruses containing double-stranded DNA that establish lifelong infections [1]. Their life cycle can be characterized as having two phases: lytic (active viral reproduction) and latency establishment (persistence of the viral genome with no particle production and the possibility of reactivation back to lytic replication) [2]. Herpesviruses derive their name from the Greek word herpes (to creep) to describe the spreading lesions along the skin manifested during herpes simplex virus or varicella zoster virus infection [3]. Referring to the quiescent nature of herpesvirus infections, latency is the hallmark of their life cycle [4]. There are over 25 viruses in the herpes family and at least eight types that infect humans, and the review focuses only on the herpesviruses that affect humans.

Herpesviruses are classified into three subfamilies: alpha, beta, and gamma- herpesviruses. Each of them is characterized differently regarding their tropism, replication cycle length, and associated diseases [4]. Alpha-herpesvirus consists of herpes simplex virus 1 (HSV-1), HSV-2, and varicella zoster virus (VZV). These viruses have a relatively fast and short replicative cycle in epithelial cells, fibroblasts, and neurons, and latency is established in sensory neurons [5]. HSV-1 and HSV-2 are among the most common human diseases, with a prevalence of 60% to 95% of the population [6]. HSV-1 is commonly associated with oral and perioral infections, and HSV-2 is known to cause genital infections. Transmission of these viruses occurs by direct contact via infected skin or secretions. HSV causes diseases ranging from mild to severe cases, such as cold sores, keratitis, corneal blindness, and encephalitis. Primary VZV infection is less common in adults, but more than 90% of them are seropositive [7]. VZV causes chickenpox in primary infection and shingles during reactivation [8,9]. The virus establishes latency within sensory ganglia [10]. Both HSV and VZV can become latent and later reactivate when individuals are immunocompromised.

Beta-herpesvirus includes human cytomegalovirus (HCMV), human herpesvirus 6A and B (HHV-6A and B), and human herpesvirus 7 (HHV-7). These viruses progress to a longer replication cycle, have species-specific tropism, and are accompanied by cell enlargement [11]. HCMV is the largest human herpesvirus with a genome of nearly 240 kb [12], and it successfully infects approximately 45 to 100% of the population worldwide [13]. HCMV causes mononucleosis-like syndrome, pneumonitis, encephalitis, bronchiolitis, retinitis, hepatitis, and gastroenteritis [14]. Primary HCMV infection can be transmitted from a seropositive organ donor, blood transfusions, infected saliva, semen, or breast milk [15]. The virus establishes lifelong latency in monocytes, dendritic cells, megakaryocytes, and progenitor myeloid cells in bone marrow. Roseolovirus (HHV-6), a common infectious agent in pediatrics, causes an exanthema subitem (*roseola infantum*) [16]. More than 80% of children by the age of 3 years are infected [17,18], and more than 90% of adults are seropositive [18]. HHV-6A is more neurotropic than HHV-6B, as DNA and mRNA are found more frequently in patients with neuroinflammatory diseases, including multiple sclerosis and rhombencephalitis [19,20]. HHV-7 has similar homology to HHV-6, which can be acquired early in life and infects more than the 90% of human population [21,22]. Moreover, HHV-7 has a narrow tissue tropism, as it infects CD4^+^ T cells, epithelial cells in the salivary glands, and skin and lung cells [23]. It may cause a roseola-like illness, acute febrile respiratory disease, and seizures in young children [19]. Both HHV-6 and HHV-7 establish persistent latency in the lymphocytes of the host and reactivate in immunocompromised hosts, especially transplant recipients [24].

Gamma-herpesvirus, including Epstein–Barr virus (EBV) and Kaposi sarcoma herpesvirus (KSHV), have very narrow tropisms regarding the host and cell type, infect lymphocytes and epithelial and endothelial cells, and establish latency in B lymphocytes [4]. EBV causes a large number of malignancies, Burkitt’s lymphoma, B-cell lymphoproliferative syndromes, nasopharyngeal carcinoma, Hodgkin’s disease, T-cell lymphomas, gastric carcinoma, and infectious mononucleosis [25,26]. KSHV causes Kaposi’s sarcoma (KS), primary effusion lymphoma (PEL), and multicentric Castleman’s disease (MCD) [27,28]. Tumor formation is observed in KS and PEL and cytokine excess in MCD. Viral latent proteins are continuously expressed, and possess the ability to inhibit apoptosis, evade the antiviral immune response, and induce proliferation [29]. Among all three herpesvirus subfamilies, for alpha- and beta-herpesviruses, the lytic cycle is selected as the default path, whereas for gamma-herpesviruses, latency predominates [4].

## 2. Replication Steps of Herpesvirus and Antiviral Targets

Herpesvirus entry is a highly complex process involving numerous viral and cellular factors [30]. Viral attachment to the host cell surface is the first step for viral entry, followed by interactions between multiple viral glycoproteins and binding receptors to facilitate capsid penetration. Each subfamily of herpesviruses has its own specific receptor-binding glycoproteins and receptors [1]. For HSV, gD is the required receptor-binding protein, which binds to Nectin-1, herpesvirus entry mediator (HVEM), and a modified form of heparan sulfate receptors, which triggers membrane fusion together with heterodimers (gH-gL) and gB while VZV lacks gD. EBV gp42 binding to human leukocyte antigen (HLA) class II is required to enter B cells, but only gH-gL and gB are sufficient for epithelial cell entry [31]. HCMV requires gH-gL together with gO in trimeric complexes before cell entry into fibroblast, epithelial and endothelial cells [32]. All herpesviruses require gH-gL as an essential component of their fusion steps of entry [1]. Then the nucleocapsid travels along microtubules to the nuclear membrane, where viral DNA is released for replication in the nucleus. Once in the nucleus, viral DNA is transcribed into mRNA by cellular RNA polymerase II. Herpesviruses are strictly regulated in three temporal cascades: immediate-early (IE), early (E), and late (L) gene expression during their productive cycles [33]. IE genes encode regulatory proteins, which are transcribed immediately by host RNA polymerase II to take control of cell defense and activate E genes. These encode necessary proteins for viral DNA replication, and the L genes mostly encode viral structural proteins. After all mRNAs are transcribed in the nucleus and translated into proteins in the cytoplasm, capsid proteins and viral DNA are packed to form new virions. Mature virions are released by exocytosis. Herpesviruses may follow similar pathways, except some proceed at slower paces than others [5].

Herpetic viral infection can be treated in many ways. Antiviral agents with inhibitory and virucidal effects including interference of viral adsorption, inhibition of the viral penetration into cells, and inhibition of viral biosynthesis and release are used to treat the infection [34]. Moreover, immunomodulators are used to boost the host immune system and induce autophagy against the virus. Antiviral drugs aim to interfere with one of the stages in any virus replication cycle, such as inhibiting DNA polymerase, reverse transcriptase, neuraminidase, and other mechanisms of actions targeted by broad-spectrum antiviral drugs [35]. The most common therapeutic drugs used in clinical practice are nucleoside analogs, which affect the DNA replication process. Acyclovir, penciclovir, valacyclovir, famciclovir, ganciclovir, and other related prodrugs are highly effective in actively replicating viruses against herpesvirus infection [36]. However, in immunocompromised patients, prolonged treatment with these drugs is more likely to develop drug-resistant strains due to mutations of thymidine nucleoside kinase or DNA polymerase. Recently, there have been some live-attenuated vaccines available for boosting cell-mediated immunity to reduce the incidence of varicella-zoster viral recurrence, but rare side effects are increasingly described due to the increased administration of VZV live vaccines worldwide [37]. No effective herpesvirus vaccine has been found to completely eradicate viral infection due to the occurrence of reactivation until now. Hence, the development of alternatively new antiviral agents with minimal side effects and reduced toxicity plays a significant role in targeting different mechanisms rather than nucleoside analogues and vaccines.

## 3. Natural Product-Derived Molecules with Anti-Herpetic Properties

Natural products and their derivatives are used to treat incurable diseases, including viral infections [38]. Many scientists from different fields are searching for new developments of safe, effective, and inexpensive antiviral agents from ethnomedicinal plants [39]. Traditional Chinese medicinal herbs have been one of the most widely used treatments for viral infectious diseases in China for many thousands of years, and research on TCM plant-derived active compounds has been investigated as viral replication inhibitors and immunomodulators [40]. Apart from plant-derived compounds, several marine natural products as well as microorganisms, fungi, and animals have also been reported for their antiherpetic activities [38]. HSV-1 and 2 are the most common cutaneous viral infections, so many scientists are interested in searching for pure compounds from botanical extracts and other natural products that exhibit strong anti-HSV activities by reducing the susceptibility of the virus both in vivo and in vitro [34,41,42,43]. On the other hand, although HHV-7 has a clear viral structure, its role in human disease is still unknown [44]. Hence, no active component is available for HHV-7. More active compounds derived from natural products may be potent antiherpetic drugs.

## 4. Methodology

A search was conducted in the databases: PubMed, Science Direct, ResearchGate, and Google Scholar for published articles. The keyword ‘herpesviruses’ was paired with ‘Chinese medicine’, ‘natural products’, ‘bioactive compounds’, ‘active components’, ‘medicinal plants’, ‘phytochemicals’, ‘alkaloids’, ‘glycosides’, ‘flavones’, ‘flavonoids’, ‘saponins’, ‘terpenes’, ‘terpenoids’, ‘phenols’, ‘polyphenols’, ‘botanical metabolites’, or ‘synthetic derivatives of natural products’ to obtain published records until August 2021. No language restriction was imposed. Data inclusion criteria included (a) studies related to derivatives of isolated compounds from natural products acting against herpesvirus, and (b) studies with natural product-inspired synthetic derivatives acting against herpesvirus. Exclusion criteria included (a) data duplication and titles or contents that did not meet the inclusion criteria, (b) reports on antiviral activities of natural products or their derivatives against the other herpesvirus, and (c) studies that involved synthetically conventional chemicals, which were not of natural origin. 

Several reviews have been published on natural products against HSV or other herpesviruses [38,42,45,46]. It may be the first review on the natural products against all herpesviruses that have affected humans since 2013 [45]. The following section describes compounds derived from plants, animals, and bacteria with potential antiherpetic activities, which have not been covered in published reviews on the same theme. These compounds are organized according to chemical structural class: alkaloids, flavonoids, terpenoids, polyphenols, anthraquinones and anthracyclines, and miscellaneous compounds. Their chemical structures are shown subsequently in Figure 1, Figure 2, Figure 3, Figure 4, Figure 5 and Figure 6, and their antiherpetic activities are listed in Table 1. The highlighted targets of natural compounds and their derivatives with several antiviral mechanisms of action are summarized in Section 6 and illustrated in Figure 7.

## 5. Compounds Showing Antiherpetic Activities

### 5.1. Alkaloids with Potential Anti-Herpetic Activities

Natural products derived from different medicinal plants were obtained and tested with low toxicity for antiherpetic activities. The chemical structures of six alkaloids possessing the antiviral activity are shown in Figure 1.

*Cephalotaxus* alkaloids have been investigated as plant secondary metabolites for over 60 years and possess diverse bioactivities including antileukemia, anticancer, antimicrobial, antiviral, immunomodulatory, and other biological effects [148]. Cephalotaxine, a benzazepine pentacyclic alkaloid, is a biologically inactive parent structure of C3-ester compounds including harringtonine (**1**) and homoharringtonine (**2**) [47]. Both active compounds isolated from *Cephalotaxus harringtonii* significantly inhibited the replication of recombinant VZV-pOka luciferase with 50% effective concentrations (EC_50_) of 9.574 and 4.654 ng/mL and 50% cytotoxic concentrations (CC_50_) in HFF cells of 45.82 and 74.71 ng/mL, respectively. Both compounds at a concentration of 10 ng/mL reduced VZV-pOka luciferase activity more effectively by 76.1% and 91.3% and induced a significant reduction in the clinical isolate VZV-YC01 DNA by 57.8% and 97.1% compared to 1.13 μg/mL acyclovir, respectively. Moreover, **2** actively inhibited HSV-1 DNA and pseudorabies virus (PRV) at 500 and 1000 nM, respectively, in Vero cells. Then, **2** and acyclovir demonstrated 50% inhibitory concentrations (IC_50_) of 139 and 789 nM against HSV-1, respectively, and **2** strongly reduced PRV glycoprotein C (gC) levels at a concentration of 500 nM [48]. 

The isoquinoline alkaloid berberine (**3**), isolated from plants including *Berberis vulgaris*, *Coptis chinensis*, *Hydrastis canadensis*, and *Rhizoma coptidis*, and its derivatives demonstrated antiviral effects against alpha-, beta-, and gamma-herpesviruses [49,50,51,52,53,54,55,56,57,58]. Then, **3** inhibited HSV-1 and 2 replications with EC_50_ values of 6.77 and 5.04 μM respectively, inhibited HSV immediate-early (IE) and early (E) gene expression, suppressed HSV-2-induced c-Jun N-terminal kinase (JNK) and NF-κB activation, and downregulated HSV-2-induced interleukin-8 (IL-8) and TNF- expression [49]. The combination index (CI) of **3** and acyclovir was 0.814, demonstrating a synergistic effect. Another study found that **3** inhibited HSV-1 and HSV-2 adsorption with 93.2 and 93.9% inhibition, respectively, and late gene products, gB and gE via Western blot analysis [50]. The IC_50_ values were 8.2 × 10^−2^ and 9.0 × 10^−2^ mg/mL against HSV-1 and 2, respectively, along with a CC_50_ of 13.2 mg/mL and a selectivity index (SI) of 147–161. No inhibitory effects were observed on viral penetration even at 10 times the concentration of IC_50_. The mechanisms targeting EBV and associated malignancies in cell culture and animal models have been reviewed [149]. Additionally, **3** demonstrated inhibitory effects of latent and lytic replication on EBV-positive NPC cells [53]. The IC_50_ was 101.3 and 56.7 µM for HONE1 cells and 124.5 and 43.1 µM for HK1-EBV cells when treated at 24 and 48 h, respectively, and at the concentration of 50 μM, **3** induced cell cycle arrest and apoptosis and reduced EBNA1 expression and the STAT3 signaling pathway in EBV-positive cells. Tumor formation was inhibited in NOD/SCID mice. Studies on EBV efficacy of **3** were investigated on tumor growth inhibition via the MAPK/ERK signaling pathway in nude mice and on reduction of cell viability via the p53-related signaling pathway targeting XAF1 and GADD45α expressions via the mitochondria-dependent pathway [55,56]. As a broad-spectrum inhibitor, **3** has antiviral properties against several HCMV strains, and **3** dose-dependently inhibited the replication of the HCMV AD-169 strain with EC_50_, CC_50_, and SI values of 2.65 μM, 390 μM, and 147, respectively, and fully inhibited the replication of the HCMV drug-resistant strain with mutations in the UL54 gene conferring cross-resistance to ganciclovir, cidofovir, foscarnet, and acyclovir [57]. Then, **3** exhibited a clear reduction early (UL44) and late (UL99) viral protein levels and efficiently inhibited viral IE-2 protein transactivating functions. Therefore, the progression of the HCMV replication cycle was impaired. 

Biliverdin (**4**), a bile pigment from chickens, has been reported for its anti-HHV-6 activity in vitro [59], and 10 μg/mL of **4** inhibited HHV-6 HST strain replication in MT-4-infected cells only during viral adsorption, but treatment with 40 μg/mL of **4** after 3 h of viral infection demonstrated no inhibitory effect. The inhibitory effects of **4** on HSV-1 or HCMV were not observed. 

Cytarabine (**5**) derived from *Cryptotheca crypta* was clinically approved by the FDA in 1969 and has been used for treating meningeal leukemias, lymphomas, and recurrent embryonal brain tumors in clinical trials [150]. In our scope of review, **5** has been used as a potent inhibitor of KSHV-induced PEL [60], and it inhibited the active KSHV replication and virion production of various PEL cells by inducing cell cycle arrest and apoptosis, inhibited host DNA and RNA syntheses, induced cellular toxicity, and induced the rapid degradation of KSHV major latent protein LANA, which is essential for replication and persistence of KSHV episomes. Additionally, 5 μM of **5** induced cell cycle arrest in BCBL1 and BC3 cells after 4 h of the treatment and treating BC1 and BCBL1 cells with the compound for 72 h decreased the intracellular KSHV DNA and LANA transcript by at least half. The 50% inhibitory concentration (IC_50_) values for JSC1, BC1, BCBL1, BC3, BCP1, and BJAB cells treated with cytarabine were 0.44, 0.49, 1.09, 1.25, 1.29, and 2.34, respectively. 

Frigocyclinone (**6**) isolated from *Streptomyces griseus* strain NTK 97 played a significant role in antibacterial and antitumor activities as well as a potent inhibitor of KSHV [61]. Molecular docking was performed to predict whether the compound possessed potential drug-like properties. Computational investigation of **6** exhibited the best inhibitory constant effect of 607.94 nM against KSHV LANA1 with an intermolecular energy and electrostatic energy of −8.89 and +0.04 Kcal/mol, respectively. The amino acid residues LYS1030, PRO1033, PHE1037, LYS1070, TRP1122, HIS 1126, and LEU1128 formed stable hydrogen bond interactions on the surface of the LANA1-frigocyclinone complex. Hence, **6** can be a good lead compound against molecular dynamics simulations, and further experiments are needed to achieve the best activity.

### 5.2. Flavonoids with Potential Anti-Herpetic Activities

Natural products derived from different medicinal plants have been obtained and tested with low toxicity for antiherpetic activities. The chemical structures of twelve flavonoids possessing antiviral activity are shown in Figure 2.

Several *Kaempferia* species, including *K. parviflora*, *K. pulchra*, and *K. galanga,* have been traditionally used for the treatment of cancer, obesity, HIV, inflammation, and other pharmaceutical activities [151]. The black rhizome extract of *Kaempferia parviflora* was found to contain 5-hydroxy-3,7-dimethoxyflavone (**7**) [62]. HCMV protease is responsible for capsid assembly, which is an essential process of viral production [152]. The IC_50_ of the Thai health-promoting herb against HCMV protease was 250 μM, and more than 70% of the enzymes was inhibited by **7** at a high concentration of 200 μg/mL. The methanolic extract inhibited HCMV protease more effectively than the aqueous extract. 

Tricin (**8**), 4′,5,7-trihydroxy-3′,5′-dimethoxyflavone, was first examined for its inhibitory effect as a novel compound against HCMV replication in the human embryonic fibroblast cell line, MRC-5 [153], and **8** was isolated from *Sasa albomarginata* (bamboo) by hot water extraction with an IC_50_ of 205 μg/mL, EC_50_ of 0.17 μg/mL, and SI of 1205.8. The study further investigated whether **8** inhibited the expression of immediate early 1 (IE1), DNA polymerase (UL54), and HCMV-induced chemokines, including CXCL11, CCL2, CCR2, and CCL5, in HCMV-infected cells [63,64,65,66]. Molecular docking revealed that **8** targeted the ATP-binding site of cyclin-dependent kinase 9 (CDK9) [67]. It dose-dependently inhibited phosphorylation at Ser2 and Ser5, resulting in inhibitory kinase activity of CDK9, and it demonstrated anti-HCMV effects with EC_50_ of 1.38 and 2.09 μM, respectively, in the plaque reduction assay. 

Baicalin (**9**), isolated from a classic traditional Chinese herbal formula, called Huanglian-Jie-Du-Tang (HLJDT) exhibited anti-ischemic stroke, neuroprotective, and antioxidative effects [154]. The anti-HSV-1 and 2 activities of **9** were found in *Plantago major* with a CC_50_ of 41.1 μg/mL and EC_50_ of more than 50 and 61.5 μg/mL, respectively [68]. Another study reported only anti-HSV-1 activity in a cytopathic effect inhibition assay with CC_50_, EC_50_, and SI values of 1000 μM, 5 μM, and 200, respectively [69]. Furthermore, **9** extracted from the root of the traditional Chinese herb *Scutellaria baicalensis* inhibited HHV-6 GS strain proliferation and adsorption with a maximum concentration of 2.8 mg/mL without any toxicity to the human T-cell strain HSB2 in vitro [70]. 

Deguelin **(10)**, [(7aS, BaS)-13, 13a-dihydro-9,10-dimethoxy-3,3-dimethyl-3H-*bis*[1] benzo-pyrano[3,4-b:6′,5′-e] pyran-7(7aH)-one, a rotenoid found in three legumes of the Fabaceae family consisting of *Lonchocarpus*, *Derris,* and *Tephrosia*, exhibited antitumorigenesis and antiproliferative activity in various cancer types in cell culture and animal models [155]. Then, **10** was first identified as the hit compound against HCMV AD169 in infected HFF cells with EC_50_, EC_90_, CC_50_, and SI values of 1.3 μM, 6.5 μM, >500 μM, and >357, respectively [71], and it blocked viral DNA synthesis and inhibited IE2-dependent transactivation of HCMV E promoters. Moreover, 250 nM **10** inhibited HCMV ganciclovir-resistant strains, lytic gene transcription, and viral E and L gene and protein expression in primary newborn human fibroblasts, NuFF-1 cells [72]. The results suggested that **10** was likely to inhibit the HCMV replication cycle from the IE to E phase [156].

Apigenin (**11**), 4′ 5 7-trihydroxyflavone, was first isolated from *Caesalpinia decapetala* and reported to have antitumor activities [157]. The compound was also found in *Ocimum basilicum* and displayed anti-herpetic activity [73]. The extract containing **11** demonstrated anti-HSV-1 and 2 with EC_50_ values of 6.7 and 9.7 mg/L, SI values of 9.0 and 6.2, and CC_50_ values of 59.9 mg/L, respectively. Compound **11** exhibited anti-HSV-1 and 2 effects by decreasing viral progeny production with EC_50_ values of 7.04 and 0.05 µg/mL in clinical strains when using chloroform as the solvent [74]. It was also inhibited with an EC_50_ of 2.33 µg/mL in the acyclovir-resistant strain. The cold lyophilized aqueous extract from the leaves of *Punica granatum* containing **11** exhibited superior anti-VZV activity compared with the ethanolic extract in inhibiting the VZV induced cytopathic effect at minimum inhibitory concentrations (MICs) of 15.625 and 31.25 μg/mL for the clinical isolate chickenpox and human epithelial diploid (HEp-2)-infecting zoster cells, respectively [75]. The results demonstrated that the extract containing **11** had the highest binding affinity against the VZV protease, which is involved in the capsid assembly and DNA packaging of the virus with an energy of −318.299 kcal moL^−1^, forming hydrogen bonds with Gly146, Leu212, Arg148, and Arg147 with a bond length between 1.9 and 3.4 Å, which suggested the potential antiviral activity of the phytochemical compound due to interaction with the VZV protease. After EBV reactivation, the IE genes Zta and Rta are expressed, and then E, L, and virion release subsequently occur. At concentrations of 20 and 50 μM, **11** completely inhibited EBV lytic proteins including Zta, Rta, EAD, and DNase, and dose-dependently blocked EBV virion production in HA and EBV-positive NPC cell lines [76]. It also inhibited EBV reactivation by suppressing EBV IE promoter Zp and Rp activities, which were induced by Zta and Rta, and demonstrated inhibitory effects against the HCMV Towne strain in colorimetric and titer reduction assays with IC_50_ values of 22 and 6.4 µM, respectively [77]. Finally, 1**1** dose-independently induced apoptosis and autophagy, activated p53, reduced ROS production, and reduced STAT3 phosphorylation in KSHV PEL cells [78].

Apigenin-7-*O*-[β-d-apiofuranosyl (1→6)-β-d-glucopyranoside] (**12**) was first isolated from the aerial parts of *Crotalaria podocarpa* using methanol extract [158], and luteolin-7-*O*-β-D-glucopyranoside (**13**) was isolated as an antioxidant from the butanol fraction of the methanol–water–acetic acid extract of *Marrubium vulgare* [159]. Later, both bioactive constituents displayed anti-herpetic activity and were found in the plant, *Lindernia crustacea* in the Scrophulariaceae family [79]. Both compounds exhibited a significant inhibitory effect on the EBV lytic protein Rta, a transcription factor during the IE stage that is required for the development of the EBV lytic cycle, at 20 μg/mL in the immunoblot analysis. The anti-EBV effect was evaluated in an EBV-containing Burkitt’s lymphoma cell line (P3HR1), and the extraction of pure compounds using ethyl acetate as a solvent exhibited the most potent anti-EBV activity.

The root extracts of *Scutellaria baicalensis* are one of the most popular traditional Chinese medicines containing wogonin (**14**), 5,7-dihydroxy-8-methoxyflavone as the major bioactive compound, and possess numerous pharmacological bioactivities, including anticancer, antiinflammation, antibacterial, and antiviral activities [160]. HSV glycoprotein D (gD) is the structural component of the viral envelope and L gene product, and the reduced levels of HSV gD suggest that **14** demonstrates anti-HSV effects [80]. The compound blocked both HSV-1 and HSV-2 viral replication by suppressing IE gene expression and interfering with E and L gene expression in vitro. NF-κB is a signaling pathway, that is required for preventing host cell apoptosis during the early stage of HSV infection, and it is an important pathway for certain pathogens to invade and infect, as well as the JNK/p38 MAPK signaling pathway. When these pathways are activated, viral mRNA transcription and DNA replication are expressed. The results showed that HSV significantly and dose-dependently inhibited HSV-induced NF-κB and JNK/p38 MAPK activation, which is an important pathway for viral replication. Hence, the compound reduced viral mRNA transcription, viral protein synthesis, and infectious virion particle titers. Moreover, 10 μg/mL **14** reduced VZV ORF4 gene expression, inhibited VZV ORF63 (IE) and ORF14 (L) genes at 48 and 72 hpi, induced type 1 interferon (IFN-1) signaling, inhibited inflammation in VZV-infected cells, and inhibited adenylate-activated protein kinase (AMPK) activity [81]. Then, **14** dose-dependently demonstrated antiproliferative effects, induced apoptosis at the G1 phase, and inhibited NF-κB activity in Raji cells. The compound also attenuated the average tumor size and weight in BALB⁄c nude mice [82]. The mice received intraperitoneal injection of **14** with a minimal toxic concentration of 8 mg/kg every 2 days for 2 weeks. The average sizes of the control group and **14**-treated group were 663.4 and 199 mm^3^, respectively, and the average tumor weight of the control group was 0.426 g compared to **14**-treated groups at 0.162 g. Hence, these results demonstrated significant inhibitory effects of tumor growth.

*Morus alba* L. known as white mulberry, was originally used in traditional Chinese medicine for treating bacterial infection, inflammation, and viral and other various diseases [83]. After purification of the root bark of the mulberry tree, the flavonoids kuwanon C (**15**), T (**16**), E (**17**), and U (**18**) were isolated. Compounds **15** and **16** demonstrated stronger anti-HSV-1 activities with IC_50_ values of 0.91 and 0.64 µg/mL and remarkably higher SI values of 230.8 and 328.1, respectively, compared to acyclovir with IC_50_ values of 1.45 µg/mL and SI 144.8. The inhibitory effects of **17** and **18** against HSV-1 and HSV-2, respectively, were comparable to acyclovir. Molecular docking studies have shown that **15**, **16**, and **18** had binding affinities of −8.2, −7.5, and −7.7 kcal/mol, respectively, with HSV-1 DNA polymerase [83]. Compound **17** was strongly bound to the active site of the HSV-2 protease with a binding affinity of −7.1 kcal/mol. These results suggested that the compounds inhibited the enzyme by forming hydrogen bonds and interacting hydrophobically with significant residues of the active site.

### 5.3. Terpenoids with Potential Anti-Herpetic Activities

Natural products derived from different medicinal plants were obtained and tested with low toxicity for antiherpetic activities. The chemical structures of seven terpenoids possessing the antiviral activity are shown in Figure 3.

The root of a Kenyan medicinal plant named *Maytenus heterophylla* containing pristimerin (**19**) was reported to have the first time for anti-HCMV activity [84]. The triterpenoid quinone methoide compound using cold diethyl ether methanol as a solvent was found to be effective against HCMV in immunocompromised hosts. The IC_50_, CC_50_, and SI were 0.53, 14.8, and 27.9 μg/mL, respectively [84]. The results demonstrated that **19** suppressed viral replications without virucidal effects on MRC-5 cells, inhibited ganciclovir (GCV)-sensitive clinical isolate 91-7S and GCV-resistant clinical isolate 93-1R strains in a dose-dependent manner, and significantly reduced IE1 and IE2 protein synthesis in MRC-5-infecting cells at concentrations of 1.3 and 6.5 μg/mL.

The root of the traditional Chinese herb, *Glycyrrhiza glabra*, containing a triterpenoid compound called glycyrrhizin (**20**) (also known as glycyrrhizic acid or glycyrrhizinic acid), and its derivatives, exerts inhibitory effects on hepatitis C virus (HCV), HSV-1, influenza virus, and severe acute respiratory syndrome associated coronaviruses (SARS-COV) [161]. Several studies have reported that glycyrrhetinic acid, the product of body metabolism from **20**, and other derivatives, demonstrated anti-HSV in vitro [85,86,87,88,162]. Then, **20** dose-dependently inhibited VZV and EBV replication with IC_50_ values of 0.71 and 0.04 mM for viral inhibition and with IC_50_ values of 21.3 and 4.8 mM for cell growth mainly at the early stage but not during adsorption [89]. The therapeutic index (TI) was 30 and 120 against VZV and EBV replication, respectively. Moreover, seven derivative compounds from **20** dose-dependently inhibited EBV infection, and three of the seven compounds containing the amino acid residues in the carbohydrate part of **20** exerted antiviral activity [90]. SUMOylation processes have been proposed as therapeutic targets of antiviral activity [91]. Firstly, **20** inhibited SUMOylation processes in LMP1-expressing cells without affecting ubiquitination processes, induced apoptosis in multiple cell lines, blocked proliferation, and reduced viral production without inducing viral reactivation. Moreover, **20** decreased KSHV latency transcription, disrupted the RNA polymerase 2 (RNAPII) complex, altered the enrichment of the RNAPII-pausing complex, and reduced the interaction between cohesion subunits, SMC3, RAD21, and SPT5 together with RNAPII. The compound abrogated RNAPII pausing at the intragenic CTCF-cohesin binding sites, resulting in a reduction in proper mRNA production and defects in sister chromatid cohesion [92]. Then, **20** demonstrated inhibitory effects against both lytic and latent KSHV in KSHV latently infected primary effusion lymphoma (PEL)-derived BC-3 cells in vitro [93]. The maximum concentration of 960 μM of the compound did not show major effects on cell viability, and the concentration of 480 μM exhibited the most potent effect on the reduction of virion DNA. It also downregulated the expression of latency-associated nuclear antigen (LANA) and upregulated viral cyclin (vCyclin) expression [94].

Triptolide (**21**), a diterpenoid triepoxide isolated from *Tripterygium wilfordii*, inhibited the proliferation of EBV-positive B lymphoma cells by downregulating latent membrane protein 1 (LMP1) transcriptional expression in BALB/c male mice [95]. The function of LMP1 has been reported to promote the proliferation of B lymphocytes, increase viral genome production, and induce viral release during lytic infection [163,164]. LMP1 promoter activity during EBV latency III infection was reduced by treatment with **21** in malignant human epithelial cervical cancer cells (HeLa) in vitro. The compound strongly reduced the growth of EBV-positive B lymphoma cells (B95-8 and P3HR-1) in BALB/c male mice as viral LMP1 mRNA expression was decreased. Moreover, **21** effectively inhibited proliferation, induced mitochondrial apoptosis, and decreased EBNA1 expression with low toxicity in EBV-positive and EBV-negative NPC cells, with IC_50_ values of 55.43, 76.56, 1.12, 11.04, and 10.66 for C666-1, HONE1/Akata, HK1/Akata, CNE1/Akata, and CNE1 cells, respectively [96]. The 0.4 mg/kg compound also inhibited tumor growth, as suggested by its weight and volume, decreased EBNA1 expression, and increased the expression of associated apoptotic proteins. In addition, 200 nM **21** decreased the hTERT mRNA, SP1 and c-Myc expression in B95-8 and P3HR-1 cells [97]. The activity of hTERT promoter and cell proliferation were inhibited by **21**, and it decreased the latency-associated nuclear antigen 1 (LANA1), which is essential for segregation, replication, and maintenance of the viral genome, and suppressed STAT3 activity and secretion of pivotal cytokines (IL-6), which plays a role in promoting the survival of KSHV-associated primary effusion lymphoma (PEL) cells [98]. Treatment with **21** dose-dependently inhibited the formation of malignant ascites and lymphomatous effusions caused by BC-3, BCBL-1, JSC-1, and BJAB cells in nonobese diabetic/severe combined immunodeficiency (NOD/SCID) mice.

Phytol (**22**), an acyclic diterpene alcohol, was one of the most significant constituents found in the essential oil of Yimucao from the herb *Leonurus japonicus* Houtt. [165]. This Chinese herb has been widely used to treat gynecological problems and possesses antibacterial activity against various gram-negative bacteria. Compound **22** has anti-inflammatory properties by downregulating the p38MAPK and NFκB signaling pathways [166]. Although this might link to our scope of review, no studies concerning herpetic activity has been conducted to confirm these pathways. Additionally, **22** isolated from *Lindernia crustacea* demonstrated an inhibitory effect of EBV lytic protein Rta at 20 μg/mL in P3HR1 cells [79].

Carvacrol (**23**), 2-methyl-5- [1-methyl ethyl]-phenol, found in the essential oils of the *Origanum vulgare*, *Satureja montana*, *Thymus vulgari*, and *Coridothymus capitatus* species, exhibited a broad spectrum of antiviral effects [167]. Compound **23** demonstrated anti-HSV-1 activity against KOS, acyclovir-resistant, and clinically isolated strains [99,102], and it might interact with the viral envelope before adsorption with an IC_50_ of 0.037% [100], and pretreatment with the virus decreased by 70% [101]. Moreover, **23** exerts anti-HSV-2 activity via inhibition of the RIP3-mediated programmed cell necrosis pathway [103]. HSV-2 infection can induce multiple cell fusions, forming polynuclear giant cells, and initiate the intracellular RIP3-mediated programmed cell necrosis pathway by promoting intracellular TNF-α, MLKL expression, and RIP3 protein activation. Additionally, **23** dose-dependently inhibited the transcriptional gene expression of ICP4, ICP27, VP16, gB, subunit of DNA polymerase (UL30), and cytokines, including RIP3, TNF-α, and MLKL. The 2% true monoterpenoid phenolic solution on HSV-2 infected BSC-1 cells demonstrated EC_50_ values of 0.19, 0.43, and 0.51 mmol/L and therapeutic index (TI) values of 9.11, 4.02, and 3.39 for treatment, prevention, and direct inactivation modes, respectively. Compound **23** also exhibited decreased levels of reverse ubiquitination in the ubiquitin-proteosome system, which was caused by HSV-2 infection.

Cypellocarpin C (**24**), the monoterpenoid ester extracted from the dried leaves of *Eucalyptus cypellocarpa*, which is commonly known as gray gum in Australia was first reported to have a structure according to spectroscopic methods to determine its antitumor-promoting effect [168]. Further study revealed that **24** was abundant in the leaves of the genus *Eucalyptus* [169]*. Eucalyptus globulus* was selected and it was found that **24** exhibited more than twice stronger anti-HSV-2 activity with an EC_50_ and SI of 0.73 µg/mL and >287.7 compared to acyclovir with an EC_50_ and SI of 1.75 µg/mL and >120 without showing cytotoxic effects as both have CC_50_ > 210 µg/mL [104]. Furthermore, tereticornate A (**25**), the triterpene ester extracted from the dried leaves of *Eucalyptus tereticornis* was first isolated and identified for its structure [170]. The anti-HSV-1 activity of **25** compared to acyclovir showed a promising result with IC_50_ values of 0.96 and 1.92 µg/mL, SI > 218.8 and >109.4, and CC_50_ > 210 µg/mL, respectively, in Vero cells. It also inhibited NF-κB activity more strongly than prednisone, which is a routinely used anti-inflammatory drug. Both compounds require further research regarding their antiherpetic activity.

### 5.4. Polyphenols with Potential Anti-Herpetic Activities

Natural products derived from different medicinal plants were obtained and tested with low toxicity for antiherpetic activities. The chemical structures of six polyphenols possessing the anti-viral activity are shown in Figure 4.

*Cis/trans*-martynoside (**26**) and *cis/trans*-isomartynoside (**27**) were first isolated from *Martynra Louisiana*, and the structures were identified using low-pressure liquid chromatography, NMR spectra, and electron impact mass spectrometry [171]. They were later isolated from *Lindernia crustacea* and displayed a significant inhibitory effect on the EBV Rta lytic cycle at 20 μg/mL via the immunoblot analysis. Using ethyl acetate to extract pure compounds showed the most potent anti-EBV activity [79].

Resveratrol (**28**), *trans*-3,5,4-trihydroxystilbene was found in several plants, including grape vines, berries, pines, pomegranates, peanuts, legumes, and soybeans with various mechanisms of action against different human and animal viruses [105,106,107,108,109,110,111,112,113,114,115,116,117,118,172]. Compound **28** and its derivatives demonstrated anti-HSV activity both in vivo and in vitro [173], it dose- and time-dependently inhibited both HSV-1 and HSV-2 replications via inhibition of ICP4 (IE gene) expression, and the inhibition of viral replication was reversible in cell culture [105]. Cream containing 12.5% and 25% **28** inhibited HSV-induced skin lesion formation as effectively as acyclovir treatment [106], and another study reported that cream containing 19% **28** inhibited HSV-induced vaginal lesion formation throughout a 9-day assay period and increased mortality rates in SKH1 mice [107]. The compound also inhibited NF–kB activation in HSV infected cells at a concentration of 219 µM, and the inhibition of viral replication was reversible and dose-dependent, which also impaired the activity of IE, E, and L genes as well as the viral DNA synthesis [108]. Oxyresveratrol, a derivative of **28**, exerted inhibitory effects against HSV-1 7401H and acyclovir-resistant strains by inhibiting the E and L phases of both HSV-1 and 2 [109]. A synergistic effect was observed when oxyresveratrol was combined with acyclovir to inhibit late protein synthesis. The IC_50_ values of oxyresveratrol were 19.8, 23.3, 23.5, 24.8, 25.5, and 21.7 µg/mL against three clinical isolates, TK-deficient and PAA-resistant HSV-1, respectively. In mice, topical treatment with oxyresveratrol delayed HSV cutaneous infection and decreased mortality rates better than oral administration. Compound **28** inhibited VZV major transcriptional regulatory protein (IE62) synthesis with EC_50_ values of 4 and 19 µM for acyclovir and **28** treatments, respectively [111]. It also inhibited 12-*O*-tetradecanoylphorbol-13-acetate (TPA)-induced Epstein–Barr early antigen activation [112], the dose-dependently reduced the EBV lytic cycle and transcription of lytic proteins, including Rta, Zta, and EA-D expression, reduced virion production in P3HR-1 cells with an EC_50_ of ~ 24 µM [113], and inhibited the EBV-induced activation of NF–kB and AP-1 [114]. Moreover, **28** induced apoptosis, inhibited the STAT3 pathway, decreased the viral LMP1 product affecting the downregulation of survivin and Mcl-1, and decreased miR-155, miR-34a, and BHRF1 expression in EBV-infected B cells [115]. Compound **28** exhibited anti-HCMV activity by blocking IE, E, and L viral proteins with an IC_50_ of 1.7 μM, CC_50_ > 400 μM, and SI ≥ 50. Compound **28** also lowered ERK1/2 activity and Egr1 expression in KSHV-infected cells, resulting in the suppression of viral reactivation from latency [117], and it induced apoptosis and autophagy, increased ROS generation without causing viral reactivation, disrupted latent viral infection, inhibited lytic gene expression, and decreased progeny virus production in PEL cells [118].

Epigallocatechin-3-gallate (**29**), a major component of the dry leaves of *Camellia sinensis*, was previously suggested to have potential antiviral activity against a broad range of viruses [122,174]. In our related scope, **29** exhibited greater antiviral activity than other green tea catechins by reducing HSV-1 and HSV-2 titers by 3,000- and 10,000-fold, respectively [119]. The compound directly affected the virion, which could bind to gB, gD, or another enveloped glycoprotein. Compound **29** is stable in the pH range found in the vagina and inactivated HSV-1 and 2 at pH 8.0 by 3 log_10_ to 4 log_10_. Theasinensin A, P2, and TF-3, digallate dimers of **29**, had antiviral activity at pH 5.7 and 8.0 and inactivated Class I, II, and III fusion proteins of enveloped viruses [120]. p-EGCG, modified from **29** inhibited infectious HSV-1 production at a concentration of 50 μM via viral adsorption in Vero cells [121]. Compound **29** also inhibited virion surface proteins and competed with heparan sulfate binding of HSV-1 [122]. MST-312 containing moieties related to **29**, dose-dependently reduced plaque formation of HSV-1 virions [123]. However, **29** exhibited virucidal activity toward HSV-1 at a concentration 80 times lower than that of MST-312, and **29** also inhibited infected cell protein 0 (ICP0), which promoted transcription of viral genes. Both compounds possessed virucidal activity against HSV virions at temperatures between 25 and 37 °C. Furthermore, **29** and acyclovir at concentrations of 25 and 50 μg/mL, respectively, increased the viability of HSV-1-induced cell death, and synergistically inhibited ICP5, TK, and gD [124]. Compound **29** inhibited the expression of EBV lytic proteins, including Rta, Zta, and EA-D, and inhibited the transcription of IE genes [125], and it dose-dependently inhibited the transcriptional activity of the BZLF1 and BMRF1 genes and suppressed the MEK/ERK1/2 and PI3-K/Akt signaling pathways involved in EBV spontaneous lytic infection in CNE1-LMP1 and B95.8 cells [126]. Compound **29** inhibited EBV spontaneous lytic infection involved in downregulation of LMP1 with an IC_50_ of 20 µM [127], and it inhibited KSHV replication by inducing cell cycle arrest in the G2-M phase and apoptosis, p53 activation, autophagy, and MAPKs via ROS generation [128]. It also reduced the production of viral progeny without causing reactivation in PEL cells. Compound **29** displayed activity against lytic KSHV infection without major effects on cell viability at 100 μM while at a concentration of 5 μM, the control group with 100% relative KSHV virion DNA was reduced to 30%.

(+)-Rutamarin (**30**), extracted from *Ruta graveolens*, has been reported to inhibit EBV lytic DNA replication [175]. Later, **30** was also found to inhibit EBV lytic DNA replication with an IC_50_ of 7.0 μM and semi-synthesis of other (−)-rutamarin derivatives, including (−)-chalepin with an IC_50_ of 69.9, and it exhibited inhibitory effects on EBV lytic DNA replication [129]. (−)-Chalepin synthesized with 5-ethyl-2-pyridine-ethanol (**4*m***), 4-pyridine-ethanol (**4*n***), and N-(2-hydroxyethyl) phthalimide (**4*p***) by a general procedure exhibited IC_50_ values of 1.5, 0.32, and 0.83 μM and SI values of 801, 211, and > 120, respectively. Moreover, **30** demonstrated human Topo II inhibition activity and blocked KSHV lytic DNA replication in BCBL-1 cells with an IC_50_ of 1.12 μM [130]. It also inhibited virion production with low toxicity, with EC_50_, CC_50_, and SI values of 1.62 μM, 94.24 μM, and 84.14, respectively.

Ginkgolic acid (**31**), a 2-hydroxy-6-alkyl benzoic acid, is an alkylphenol constituent of the leaves and fruits of *Ginkgo Biloba*, and it is most experimentally used to date C15:1. An extract containing < 5 ppm **31** demonstrated anti-HSV-1 and -2 activities before viral adsorption to the cell surface, disrupted the viral structure, and inhibited HSV-1 virion entry without cytotoxicity [131]. At a concentration of 50 μM, **31** blocked downstream HSV-1 protein synthesis, such as the production of HSV-1 immediate early (ICP27), early (ICP8), and late (US11) proteins [133]. It significantly reduced mortality, infection score, and durations of HSV-1 cutaneous infection in a zosteriform model in BALB/cJ mice in vivo and inhibited an acyclovir-resistant strain of HSV-1 in Vero cells [132]. Moreover, **31** inhibited HCMV DNA synthesis in a dose-dependent manner and prevented plaque formation. The results were determined by decreasing viral DNA copies of HCMV clinical strains (CH19 and BI6) in human foreskin fibroblast (HFF) cells.

### 5.5. Anthraquinones and Anthracyclines with Potential Antiherpetic Activities

Natural products derived from different medicinal plants were obtained and tested with low toxicity for antiherpetic activities. The chemical structures of two anthraquinones and three anthracyclines possessing antiviral activity are shown in Figure 5.

Anthraquinones are a class of aromatic compounds demonstrating a broad spectrum of bioactivities. Emodin (**32**) and aloe-emodin (**33**) are active ingredients in different Chinese herbs including *Aloe vera*, *Cassia occidentalis*, *Rheum palmatum,* and *Polygonam multiflorumto* and possesses broad pharmacological effects, such as antibacterial, antiviral, antiinflammatory, anticancer, neuroprotective, hepatoprotective, and anticardiovascular effects [134,135,136,137,138,139,176,177]. At a concentration of 50 μg/mL, **32** inhibited HSV-1 and 2 in Hep-2 cells with antiviral indices of 2.07 and 3.53, respectively, and the compound with equivalent antiviral efficacy to acyclovir also increased the survival rate and time of HSV-infected mice with higher efficacy in eliminating HSV-infected brain, heart, liver, and ganglion compared to the viral control [135]. It also decreased the TLR3 pathway and its downstream molecules, trif, TRADD, traf3, TRAF6, Nemo, p38, and IRF3, as well as f IL-6, TNF-α and IFN-β expression in HSV-infected brain tissues of mice [136]. Furthermore, **32** at a concentration of 4.2 μg/mL completely inhibited the transcription of BRLF1 and BZLF1 mRNA expression in EBV lytic DNA replication with an EC_50_ of 1.2 μg/mL and dose-dependently decreased lytic protein expressions including Rta, Zta, and EA-D [137]. It further blocked virion production, repressed Zta and Rta promoter activities, suppressed SP1 expression and tumorigenic properties with CC_50_ values of 31, 58, 65, 79 μM for Tw01-, HONE-1-, HA-, NA-, and EBV- positive epithelial cell lines, respectively, and suppressed EBV-induced tumor growth in mice [138]. Compound **32** demonstrated both an anti-HCMV AD-169 strain with EC_50_, IC_50_, and TI values of 4.1 μM, 9.6 μM, and 2.3, respectively, and a ganciclovir-resistant strain with EC_50_, IC_50_, and TI values of 3.7 μM, 12.6 μM, and 3.4, respectively [139]. Compound **33** also demonstrated an anti-HCMV AD-169 strain with an EC_50_ and IC_50_ of > 37.0 μM. Both **32** and **33**, which were isolated from *Lindernia crustacea* demonstrated an inhibitory effect of EBV lytic protein Rta at 20 μg/mL in infected P3HR1 cells [79]. Both pure compounds were extracted with ethyl acetate. The EC_50_ and C_50_ were ~29 μg/mL and 84.56, respectively, when treated with **32** on EBV infected P3HR1 cells, which was required to inhibit EBV IE protein expression.

Anthracyclines are a class of drugs produced by different wild type *Streptomyces bacterium* strains, used in cancer chemotherapy [178]. Three FDA-approved drugs, daunorubicin (**34**), doxorubicin (**35**), and epirubicin (**36**), are topoisomerase II inhibitors that are capable of intercalating DNA and inducing cytotoxic effects in cancer cells [140]. At concentration of 10 μM, all three compounds induced effective luciferase expression under control of the PAN or RTA promoters of KSHV infecting BCBL1 and Vero cells. All three compounds demonstrated efficient lytic induction of KSHV compared with sodium butyrate.

### 5.6. Miscellaneous Compounds with Potential Antiherpetic Activities

Natural products derived from different medicinal plants have been obtained and tested with low toxicity for antiherpetic activities. The chemical structures of twelve miscellaneous compounds possessing antiviral activity are shown in Figure 6.

Allicin (**37**) from fresh garlic extract displayed activity against a wide range of viruses, including enveloped and non-enveloped viruses [93,179]. The antiviral activity of compound **37** and its derivatives was pre-clinical and clinically investigated in a previous review [180]. In our review, we found that compound **37** exhibited antiviral activity against both latent and lytic KSHV, but the former was more effective than the latter. Compound **37** at a concentration of 15 μM significantly reduced the virion DNA in KSHV. The compound demonstrated maximum tolerable drug concentrations without major effects on cell viability at a very low concentration of 6 μM.

Egonol and homoegonol isolated from *Styrax officinalis* and artemisinin, an enantiomerically pure sesquiterpene lactone isolated from *Artemisia annua*, demonstrated significant bioactivities, such as anti-inflammatory, antioxidant, antimicrobial, anti-cancer, and other inhibitory mechanisms of action [181,182,183,184,185,186]. The hybridization of natural products was reported to have antimalarial, anticancer, and antiviral activity. Within our scope of antiherpetic activities, artemisinin-egonol (**38**) and artemisinin-homoegonol hybrids (**39**) were obtained via etherification reactions [141,187]. They exhibited strong anti-HCMV activity with EC_50_ values of 0.17 and 0.13 μM, respectively [188]. Compound **33** has 20-fold higher activity against the HCMV AD169-GFP strain infecting human primary fibroblasts than ganciclovir, which has an EC_50_ of 2.60 μM.

Byzantionoside B (**40**) was first isolated from *Stachys byzantina* as the C-9 epimer of blumenol C glucoside [189], and the absolute configurations have been determined [190]. The compound was later found in *Lindernia crustacea* and demonstrated significant inhibitory effect on the EBV lytic cycle at 20 μg/mL in the immunoblot analysis. Compound **40** was extracted using ethyl acetate to exhibit the most potent anti-EBV activity [79].

*Hypericum* species have been phytochemically and pharmacologically investigated with various bioactivities, including cytotoxicity against cancer cell lines, anticancer, antibacterial, antioxidant, anti-inflammatory, and antiviral activities [191]. Highlighted compounds were isolated from *Hypericum japonicum* with anti-EBV and KSHV activity, and the structures were confirmed by extensive NMR spectroscopic data and calculated ECD analyses. (+)-Hyperjaponicol B (**41**) and hyperjaponicol D (**42**) inhibited EBV DNA replication in B95-8 cells with EC_50_ values of 0.57 and 0.49 μM and SI > 52.63 and 106.78, respectively, and both compounds exhibited ~5-fold more efficacy than ganciclovir with an EC_50_ value of 2.86 μM and an SI value of 104.50 [142]. Hyperjaponicol H (**43**) moderately inhibited EBV lytic DNA replication in B95-8 cells with an EC_50_ of 25.00 µM, CC_50_ > 50 µM, and SI > 2 [143]. (+)-Japonicol B (**44**) moderately exhibited anti-KSHV activities with EC_50_, CC_50_, and SI values of 8.75 μM, 140.60 μM, and 16.06, respectively [144]. (+)-japonicols E (**45**) and H (**46**), two acylphloroglucinol-based meroterpenoids isolated in the form of white amorphous powder and reddish-brown oil exhibited strong inhibitory effects against KSHV lytic replication [145]. The IC_50_ values of (**45**) and (**46**) were 8.30 and 4.90 μM, the SI values were 23.49 and 25.70, respectively. (+)-japonone A (**47**) exhibited inhibitory activity with an IC_50_ of 166.0 μM, CC_50_ > 500 μM, and SI > 3.01 on KSHV lytic replication with less toxicity and better selectivity than (−)-japonone A with inert activities on anti-KSHV [146]. Japopyrone B (**48**) exhibited potential inhibitory efficacy on TPA-induced KSHV lytic replication with an IC_50_ of 29.46 µM, CC_50_ > 200, and SI > 6.79 [147].

## 6. Mechanisms of Action

Natural products can target different stages of herpesviral life cycles, which are summarized and shown in Figure 7.

### 6.1. Blocking Viral Adsorption and Entry

During viral attachment, herpesviral entry is mediated by the interaction of multiple glycoproteins and binding receptors on the cell surface to facilitate capsid penetration. Glycoprotein C (gC) plays a vital role in terms of viral attachment in HSV by interacting with heparan sulfate carbohydrates on the host cell surface [90]. Glycoprotein D (gD) binds to Nectin-1, herpesvirus entry mediator (HVEM), and a modified form of heparan sulfate receptors, which triggers membrane fusion together with heterodimers (gH-gL) and gB [1]. Viral glycoproteins could be vulnerable to mutations causing resistance to viral entry by altering interactions with their receptors [91]. Natural compounds have been proven to have better inhibitory effects than conventional drugs without showing major cytotoxic effects in blocking viral adsorption and entry from drug-resistant strains in vitro. Compounds, including berberine (**3**), biliverdin (**4**), baicalin (**9**), and epigallocatechin-3-gallate (**29**), are described in Section 5. The possible mechanism of action of these compounds showed in Figure 7. Further investigation is required to clearly understand the mechanisms of action.

### 6.2. Inhibition of Viral Replication

The Viral enzyme DNA-polymerase is the major target of many drug-resistant strains in herpesviruses. Numerous antiviral compounds from natural sources can be effectively used against viral DNA replication without resistance in vitro. Cytarabine (**5**), tricin (**8**), carvacrol (**23**), (+)-rutamarin (**30**), (+)-hyperjaponicol B (**41**), hyperjaponicol D (**42**), and hyperjaponicol H (**43**) are described in Section 5. The possible mechanisms of action of these compounds are shown in Figure 7. Further investigation is required to clearly understand the mechanisms of action.

### 6.3. Inhibition of NF-κB Activity

An inactive form of nuclear factor kappa-light-chain-enhancer of activated B cells (NF-κB) dimers is localized in the cytoplasm by binding to inhibitors of κB (IκB), and after degradation of this inhibitory protein causes translocation (during early stage of infection) of the dimers into the nucleus, NF-κB regulates the expression of numerous genes involved in apoptotic, inflammatory, and immune responses [101]. NF-κB also induces the expression of various proinflammatory genes, including cytokines and chemokines [102]. The activation of NF-κB synthesizes numerous antiapoptotic factors, which are required for preventing host cell apoptosis [103]. Based on the evidence, the compound inhibiting NF-κB activation or translocation may interfere with viral replication via programmed cell death, including berberine (**3**), wogonin (**14**), and resveratrol (**28**), which are described in Section 5. The possible mechanisms of action of these compounds are shown in Figure 7. Further investigation is required to clearly understand the mechanisms of action.

### 6.4. Compounds Affecting Viral Replication by Other Mechanisms

Natural compounds affect antiviral activity via different pathways. Berberine (**3**) inhibited tumor growth via the MAPK/ERK signaling pathway in nude mice and via the p53-related signaling pathway and its downstream molecules targeting XAF1 and GADD45α expression via a mitochondria-dependent pathway [55,56]. Glycyrrhizin (**20**) inhibited small ubiquitin-like modifier (SUMO)-ylating processes as a novel therapeutic target of antiviral activity [91]. During the latency of EBV infection, LMP1 increased protein SUMOylation, aiding tumorigenesis. Then, **20** blocked cell proliferation, increased cell death without inducing viral reactivation, and impeded viral production in infected cells. Carvacrol (**23**) exerted anti-HSV-2 activity via inhibition of the RIP3-mediated programmed cell necrosis pathway and exhibited decreased ubiquitination levels in the ubiquitin-proteosome system, which was caused by HSV-2 infection [102]. Toll-like receptor 3 (TLR3) plays a significant role in immune responses to herpesviruses. After viral stimulation, TLR3 subsequently transmitted signals to TAK1 via TRAF6, TRADD, p38, and NF-κB, which produced the expression of inflammatory factors [136]. Emodin (**32**) suppressed the TLR3 pathway and its downstream molecules, trif, TRADD, traf3, TRAF6, Nemo, p38, and IRF3, as well as f IL-6, TNF-α, and IFN-β expression in HSV-infected brain tissues of mice [136]. Further investigation is required to clearly understand the mechanisms of action.

### 6.5. Efficacy of Natural Compounds In Vivo

Numerous natural compounds and extracts with antiherpetic activity have been analyzed in vitro. These results can be further investigated to evaluate their efficacy in living organisms via in vivo studies. However, only berberine (**3**), wogonin (**14**), triptolide (**21**), resveratrol (**28**), ginkgolic acid (**31**), and emodin (**32**) have been tested in both cell culture experiments and animal studies. The results are summarized in Table 1 and discussed in Section 5. Further investigation is required to clearly understand their mechanisms of action.

## 7. Summary and Additional Comment

Nucleoside analog drugs, including acyclovir, ganciclovir, valaciclovir, and other related prodrugs, are used as the current standard therapy for herpesvirus infection. However, after frequent and long-term treatment, drug resistance and undesirable effects occur that limit the available treatment options especially in immunocompromised patients. Hence, the search for newly developed, safe, and effective antiviral drugs is clearly needed. Most medicinal plants are promising sources of new antiviral compounds. In this review, natural compounds isolated from plants, animals, fungi, and bacteria are summarized with different antiviral mechanisms affecting the herpesviral surface, viral adsorption, entry, replication, capsid assembly, and release. The compounds are classified into alkaloids, flavonoids, terpenoids, polyphenols, anthraquinones, anthracyclines, and miscellaneous compounds. Compounds, including the alkaloid berberine (**3**), the flavonoid deguelin (**10**), the terpenoid pristimerin (**19**), the phenolic ginkgolic acid (**31**), and the polysaccharide Cucumis melo sulfated pectin (SPCm), exhibited better antiherpetic activity than conventional drugs in vitro [57,72,84,132,192]. These candidates were further investigated for alternative treatment of drug-resistant viruses. For example, **3**, **10**, and **19** can effectively inhibit ganciclovir-resistant strain in HCMV [57,72,84]: **3** fully inhibited the HCMV replication in ganciclovir-resistant strain due to a mutation in a DNA polymerase catalytic subunit (UL54 gene); **10** can also inhibit the drug-resistant HCMV strain at nanomolar concentration; and **19** can suppress both the ganciclovir-resistant (93-1R) and sensitive strains (91-7S) in HCMV replication. Both **31** and SPCm inhibited an acyclovir-resistant HSV-1 strain (17+ and AR-29, respectively), but the latter also inhibited the acyclovir-sensitive HSV-1 strain (KOS).

A few compounds, including berberine (**3**), wogonin (**14**), triptolide (**21**), resveratrol (**28**), ginkgolic acid (**31**), and emodin (**32**), have been evaluated for their efficacy against viral infection in vivo. For example, **3** inhibited tumor formation in NOD/SCID mice via the MAPK/ERK signaling pathway and reduced cell viability through the p53-related signaling pathway by targeting XAF1 and GADD45α expression via a mitochondria-dependent pathway [55,56]. Treatment with **14** attenuated the average tumor size and weight in BALB⁄c nude mice [82]. Then, **21** can efficiently reduce the growth of B95-8-induced B lymphoma and downregulate LANA1 expression, and viral replication in PEL cells in mice [95,96,98]. Topical application of **28** inhibited HSV-induced skin lesions and increased survivability in mice [106,107]. The mortality rate was decreased by treatment with **31**, and the overall infection score and average time to death of HSV-1 cutaneous infection in female BALB/cJ mice was reduced comparing to the DMSO control group in both HEC gel and PEG [132]. Finally, **32** suppressed HSV-infected brain, heart, liver, and ganglion tissues, increased survival rate, and inhibited EBV-induced tumor growth in mice [135,136,138]. These results show that these compounds can be potential candidates for future clinical trials, and further investigations are required to compare these compounds with the standard therapy of herpesviral infection in clinical trials and in vivo studies.

## 8. Conclusions

Human herpesvirus is known to establish lifelong latent infection. Until now, no prophylactic herpesvirus vaccine has been found to be effective and completely eradicated the viral infection. In this review, we have summarized the findings of previous literature regarding natural compounds derived from medicinal plants, animals, and bacteria with promising antiviral activity against herpesvirus infections in both cell culture and animal models. Most natural compounds are briefly demonstrated as safe and effective treatments to inhibit different types of herpesviruses, in which the IC_50_, EC_50_, CC_50_, and SI are shown in Table 1. These natural compounds can target different stages of herpesviral life cycles, which are shown in Figure 7 and discussed in Section 6. For instance, tricin (**8**) dose-dependently inhibited the expression of immediate early 2 mRNA, 50 μM of ginkgolic acid (**31**) blocked downstream HSV-1 protein synthesis by inhibiting viral entry, which is involved in the production of HSV-1 immediate early (ICP27), early (ICP8), and late (US11) proteins, and treatment with 250 nM deguelin (**10**) significantly reduced the expression of the viral early (pUL44) and late (IE2-60) proteins. Overall, most natural compounds in the review have been studied in vitro with various mechanisms of action based on different screening methods, which are comparatively better than nucleoside analogs. Only a few studies have investigated them in animal models. The evaluations for promising antiherpetic compounds in animal trials and further clinical trials as alternative therapies against herpesvirus infections in humans would be still needed in the future.

## Figures and Tables

**Figure 1 molecules-26-06290-f001:**
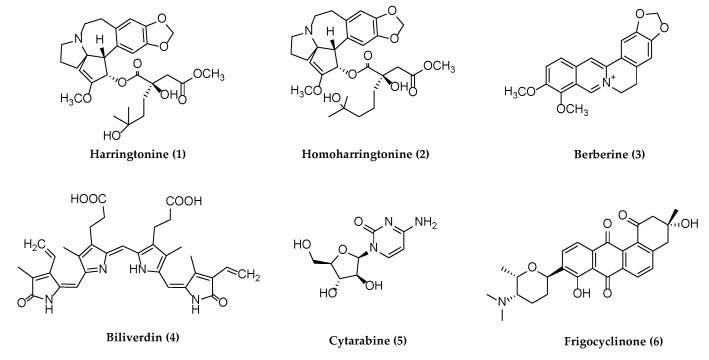
Alkaloids with potential anti-herpetic activities.

**Figure 2 molecules-26-06290-f002:**
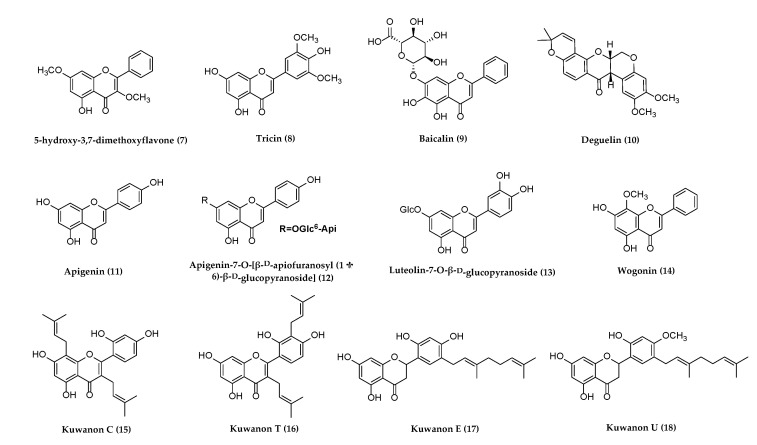
Flavonoids with potential anti-herpetic activities.

**Figure 3 molecules-26-06290-f003:**
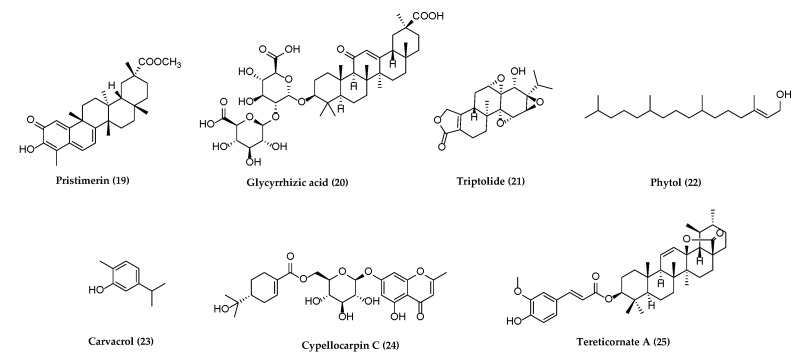
Terpenoids with potential anti-herpetic activities.

**Figure 4 molecules-26-06290-f004:**
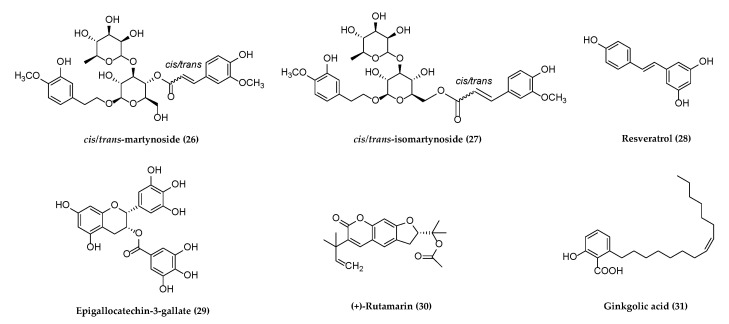
Polyphenols with potential antiherpetic activities.

**Figure 5 molecules-26-06290-f005:**
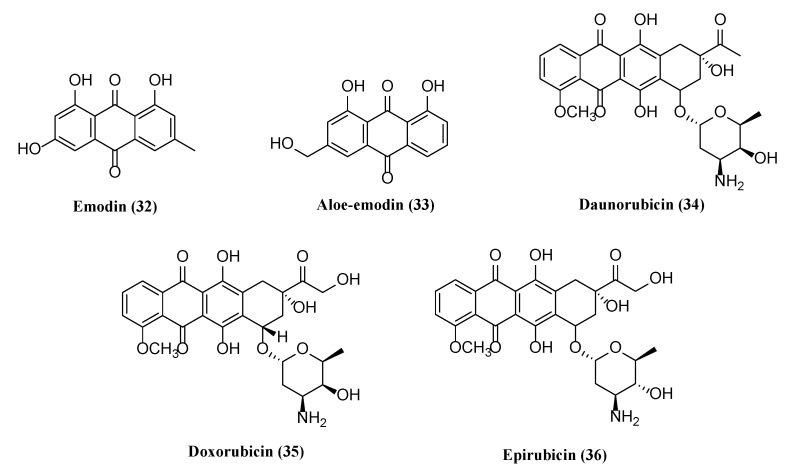
Anthraquinones and anthracyclines with potential antiherpetic activities.

**Figure 6 molecules-26-06290-f006:**
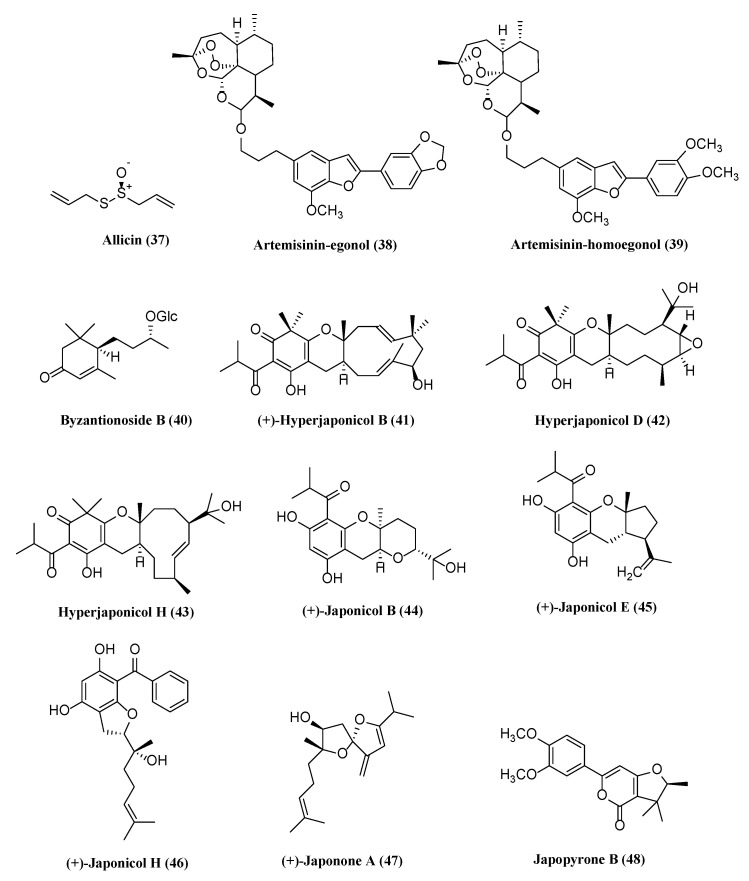
Miscellaneous with potential antiherpetic activities.

**Figure 7 molecules-26-06290-f007:**
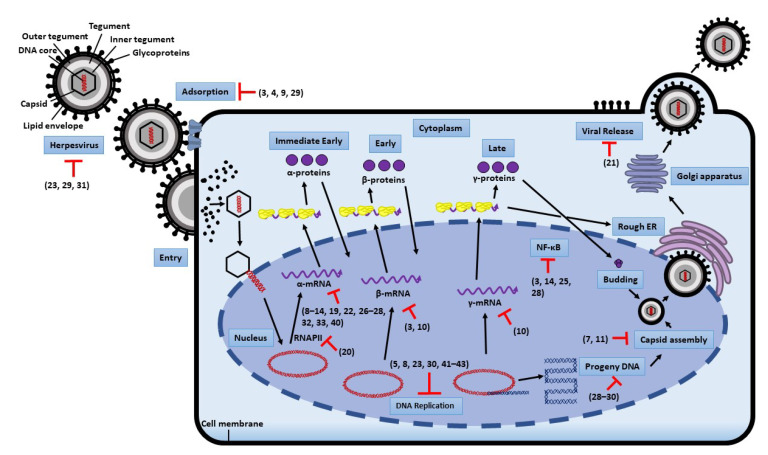
Herpesvirus lytic replication contains highly coordinated phases of viral genetic infection, which are expressed in a synchronized cascade. Immediate early (IE) genes, which are proteins involved in immune evasion and facilitating early (E) gene expression, are transcribed first. The E proteins facilitate viral DNA replication, and once completed, they trigger the late (L) gene expression, in which the viral component facilitates viral egress. If any stages are inhibited, the following stages are impacted. Different natural antiviral compounds are expressed in number.

**Table 1 molecules-26-06290-t001:** Natural products derived from plants, animals, and bacteria demonstrating antiherpetic activities.

No.	Compound	Origin	Virus (Strain)	Cell Line/Animal	EC_50_/IC50; SI, CC_50_	Mode of Action	Reference
**1**	Harringtonine (**1**)	*Cephalotaxus harringtonii*	VZV (pOka-luciferase, YC01)	HFF	EC_50_: 9.574 and 4.654 ng/mL of **1** and **2**, CC_50_ = 45.82 and 74.71 ng/mL, respectively.	Induced down-regulation of VZV lytic genes by **1** and **2**.	[47]
**2**	Homoharringtonine (**2**)		HSV-1 (F); PRV (Fa)	Vero, HEK293T, HeLa	IC_50_: 139 and 789 nM of **2** and acyclovir respectively in HSV-1.	Antagonizes the phosphorylation level of endogenous and exogenous eIF4E.	[48]
**3**	Berberine (**3**)	*Berberis vulgaris*	HSV-1 (HF, Blue); HSV-2 (G)	HEK293T, Vero, HEC-1-A	EC_50_: 6.77, 5.04 μM of **3** in HSV-1, HSV-2, respectively.	Modulating cellular JNK and NF-κB pathways.	[49]
			HSV-1 (F); HSV-2 (333)	Vero	IC_50_: 8.2 × 10^−2^ and 9.0 × 10^−2^ mg/mL of **3** in HSV-1 and HSV-2, respectively; SI = 147–161, CC_50_ = 13.2 mg/mL.	Inhibited HSV-1 and HSV-2 adsorption, gB, gE via western blot analysis.	[50]
			HSV-2 (333)	VK2/E6E7	-	Weak anti-HSV-2 activity	[51]
			HSV (GFP)	RAW264.7	-	Induction of antiviral state via type I IFN stimulation.	[52]
			EBV	HONE1, HK1, NOD/SCID mice	IC_50_: 101.3 and 56.7 µM for HONE1 cells, IC_50_ = 124.5 and 43.1 µM for HK1-EBV cells when treated at 24, 48 h, respectively.	Decreases the expression of EBNA1.	[53]
				HK1, HONE1, C666-1, NP460, Nude mice	IC_50_: ~ 100 μM for HONE1. IC_50_ ~ 400 μM for HK1, C666-1, NP460 NPC cell line.	Inhibition of STAT3 activation in NPC cells.	[54]
				CNE2, BALB/C-NU male mice	IC_50_: 21.71 μM	Regulated the expression of key protein in MAPK/ERK pathway by **3** combined with Rg3.	[55]
				IM-9	-	Upregulation of XAF1 and GADD45α expression by MAPK and functional p53.	[56]
			HCMV (AD169, clinical isolates, drug-resistant)	HFF, NIH 3T3, HELF, U373-MG	EC_50_: 2.65 μM; SI: 147, CC_50_: 390 μM.	Interference with the transactivating activity of viral IE2 protein.	[57]
			HCMV	MRC-5	IC_50_: 0.68 and 0.91 μM of **3** and ganciclovir, respectively.	Berberine Chloride inhibited HCMV replication.	[58]
**4**	Biliverdin (**4**)	A chicken bile pigment	HHV-6 (HST)	MT-4	-	Anti-HHV-6 activity in vitro during viral adsorption.	[59]
**5**	Cytarabine (**5**)	*Cryptotheca crypta*	KSHV	BCBL1, BC3, JSC1, BCP1, BJAB	-	Inducing cell cycle arrest, apoptosis, rapid degradation of KSHV LANA.	[60]
**6**	Frigocyclinone (**6**)	*Streptomyces griseus strain NTK 97*	KSHV	-	-	Inhibited KSHV LANA1 via computational investigation.	[61]
**7**	5-hydroxy-3,7- dimethoxyflavone (**7**)	*Kaempferia* *parviflora*	HCMV (AD169)	Serine protease	IC_50_: 250 μM	Inhibited HCMV protease.	[62]
**8**	Tricin (**8**)	*Sasa albomarginata*	HCMV (AD169)	MRC-5	EC_50_: 0.17 μg/mL/IC_50_: 205 μg/mL; SI: 1205.8.	Anti-HCMV activity in vitro.	[63]
			HCMV (Towne)	MRC-5	-	Inhibited CXCL11 mRNA expression of IE and/or E stages.	[64]
				HEL	-	Depressing CCL2 expression.	[65]
					-	Inhibits the CCL5 induction.	[66]
					EC_50_: 2.09 μM/IC_50_: 1.38 μM.	Inhibited kinase activity of CDK9.	[67]
**9**	Baicalin (**9**)	*Scutellariae baicalensis*	HSV-1 (KOS); HSV-2 (196)	BCC-1/KMC	EC_50_ > 50 and 61.5 μg/mL in HSV-1 and HSV-2, respectively, CC_50_ = 41.1 μg/mL.	Anti-HSV-1 and HSV-2 activities.	[68]
			HSV-1 (KOS)	Vero	EC_50_: 5 μM; SI: 200, CC_50_: 1000 μM.	Anti-HSV-1 activity involved the intracellular effect.	[69]
			HHV-6 (GS)	HSB2	-	Inhibited viral proliferation and adsorption.	[70]
**10**	Deguelin (**10**)	*Derris trifoliata*	HCMV (AD169)	HFF	EC_50_ and EC_90_: 1.3 and 6.5 μM respectively, CC_50_ > 500, SI > 357.	Inhibited the viral transcription factor Immediate-Early 2 (IE2).	[71]
			HCMV (Ganciclovir-resistant)	NuFF-1	-	Inhibits HCMV lytic replication.	[72]
**11**	Apigenin (**11**)	*Punica granatum*	HSV-1 (KOS); HSV-2 (196)	BCC-1/KMC	EC_50_: 6.7 and 9.7 mg/L, SI: 9.0 and 6.2 respectively. Both CC_50_: 59.9 mg/L.	Anti-HSV-1 and HSV-2 activities.	[73]
			HSV-1 (KOS)	Vero	CC_50_: 250 μM, EC_50_: 5 μM, and SI: 50.	Anti-HSV-1 activity.	[69]
			HSV-1 (Clinical isolates); HSV-2 (Clinical isolates, acyclovir-resistant)		EC_50_: 7.04 and 0.05 µg/mL respectively. EC_50_: 2.33 µg/mL in acyclovir-resistant strain.	Reduced viral progeny production; interfered with cell-to-cell virus spread.	[74]
			VZV (Clinical isolates)	Hep-2	-	Interact with the protease of HHV-3 through discovery studio.	[75]
			EBV	NA, HA, P3HR1	-	Inhibits EBV reactivation by suppressing the promoter activities of two viral IE genes.	[76]
			HCMV (Towne)	HEL 299	IC_50_: 22 and 6.4 µM.	Inhibitory effect against HCMV.	[77]
			KSHV	BC3, BCBL1, B cells	-	Modulate pro-apoptotic and pro-survival pathways.	[78]
**12**	Apigenin-7-*O*-[β-d-apiofuranosyl (1 → 6)-β-D-glucopyranoside] (**12**)	*Lindernia* *crustacea*	EBV	P3HR-1	-	Significant inhibitory effect on the EBV Rta lytic cycle.	[79]
**13**	Luteolin-7-*O*-β-d-glucopyranoside (**13**)				-		
**14**	Wogonin (**14**)	*Scutellaria baicalensis*	HSV-1 (HF); HSV-2 (G)	Vero, HEC-1-A	-	Modulating cellular NF-κB and MAPK pathways.	[80]
			VZV (YC01, YC03)	HFF	-	Modulation of type I interferon signaling and adenosine monophosphate-activated protein kinase activity.	[81]
			EBV	Raji, four-week-old male BALB/c nude mice	**-**	Downregulating the expression of NF-κB through LMP1/miR-155/NF-κB/PU.1 pathway.	[82]
**15**	Kuwanon C (**15**)	*Morus alba*	HSV-1 (KOS)	Vero	IC_50_: 0.91 and 1.45 µg/mL; SI: 230.8 and 144.8 for **15** and acyclovir, respectively, CC_50_ > 210 µg/mL.	Molecular docking: targeting HSV-1 DNA polymerase and HSV-2 protease.	[83]
**16**	Kuwanon T (**16**)		HSV-1 (KOS)		IC_50_: 0.64 and 1.45 µg/mL; SI 238.1 and 144.8 for **16** and acyclovir, respectively, CC_50_ > 210 µg/mL.	Molecular docking: targeting HSV-1 DNA polymerase and HSV-2 protease.	
**17**	Kuwanon E (**17**)		HSV-2 (Clinical isolates)		EC_50_: 1.61 and 1.65 µg/mL; SI 130.4 and 127.3 for **17** and acyclovir, respectively, CC_50_ > 210 µg/mL.	Molecular docking: targeting HSV-1 DNA polymerase and HSV-2 protease.	
**18**	Kuwanon U (**18**)		HSV-1 (KOS)		IC_50_: 1.93 and 1.45 µg/mL; SI 108.8 and 144.8 for **18** and acyclovir, respectively, CC_50_ > 210 µg/mL.	Molecular docking: targeting HSV-1 DNA polymerase and HSV-2 protease.	
**19**	Pristimerin (**19**)	*Maytenus heterophylla*	HCMV (93-1R, 91-7S)	MRC-5	IC_50_: 0.53 μg/mL; SI: 27.9, CC_50_:14.8 μg/ml.	Inhibited viral replication.	[84]
**20**	Glycyrrhizin (**20**)	*Glycyrrhiza glabra*	HSV-1	HEp-2	-	Inhibitory effect on the growth of HSV-1.	[85]
			HSV-1 (17 syn+); HSV-2 (HG52)	BHK, Vero	-	Reduced of HSV-1 and HSV-2 infectious virus.	[86]
			HSV-1 (KOS)	HEF	IC_50_: 3.6 µM.	Inhibitory effect on HSV-1 replication.	[87]
				Vero	IC_50_: 225 µM; SI > 2.7 CC_50_ > 608 µM.	Anti-HSV-1 activity.	[88]
			VZV, EBV	Raji, P3HR-1	IC_50_: 0.71 and 0.04 mM for viral inhibition, IC_50_ = 21.3 and 4.8 mM for cell growth, TI: 30 and 120, respectively.	Inhibited VZV and EBV replication mainly at the early stage.	[89]
			EBV		-	Inhibited EBV infection.	[90]
				HEK	**-**	Targeted the first step of the sumoylation process.	[91]
			KSHV	BC-3, BCBL-1, BCP-1, BC-1, BC-2, keratinocytes, CB33, Ramos, SLK, KS2616, HUVEC, BJAB	**-**	Downregulating LANA, upregulating vCyclin, and inducing cell death in KSHV-infected cells.	[92]
				BCBL-1, 293T	**-**	Disruption of CTCF-cohesin-mediated RNA polymerase II pausing and sister chromatid cohesion.	[93]
				BC-3	-	Showed antiviral activity against both latent and lytic KSHV.	[94]
**21**	Triptolide (**21**)	*Tripterygium wilfordii*	EBV	B95-8 P3HR-1 HONE1/Akata C666-1 293T HeLa, BALB/c male mice	**-**	Reduced LMP1 expression in EBV-positive B lymphocytes.	[95]
				HONE1/Akata, HK1/Akata, C666-1 CNE1/Akata, CNE1, BALB/c male mice	IC_50_: 55.43, 76.56, 1.12, 11.04, 10.66 for C666-1, HONE1/Akata, HK1/Akata, CNE1/Akata and CNE1 cells.	Increasing sensitivity of mitochondria apoptosis of nasopharyngeal carcinoma cells.	[96]
				B95-8, P3HR-1, 293T	-	Downregulating translation factors SP1 and c-Myc.	[97]
			KSHV	BCBL-1, JSC-1, BC-3, BJAB, P3HR-1, NOD/SCID mice	**-**	Downregulated LANA1 expression, reduced viral titers.	[98]
**22**	Phytol (**22**)	*Lindernia crustacea*	EBV	P3HR-1	**-**	Inhibited Rta expression.	[79]
**23**	Carvacrol (**23**)	*Lippia graveolens*	HSV-1 (KOS, acyclovir-resistant)	HEp-2	CC_50_: 250 µg/mL, EC_50_: 48.6 and 28.6 µg/ mL against KOS and acyclovir-resistant strain, respectively.	Anti-HSV-1 activity.	[99]
			HSV-1	Vero	IC_50_: 0.037%.	Interact with viral envelope before the adsorption.	[100]
			HSV-1 (KOS)		**-**	70% decrease in pretreatment of virus.	[101]
			HSV-1 (Clinical isolates)		IC_50_: 7 µM (1.05 µg/mL), CC_50_: 300 µM (45 µg/mL), SI: 43.	Anti-HSV-1 activity.	[102]
			HSV-2 (G)	BSC-1	**-**	Inhibited HSV-2 induced RIP3-mediated programmed cell necrosis pathway and ubiquitin-proteasome system.	[103]
**24**	Cypellocarpin C (**24**)	*Eucalyptus globulus*	HSV-2 (Clinical Isolates)	Vero	EC_50_: 0.73 and 1.75 µg/mL; SI > 287.7 and >120 of **24** and acyclovir, respectively, CC_50_ > 210 µg/mL.	Stronger anti-HSV-2 compared to acyclovir.	[104]
**25**	Tereticornate A (**25**)		HSV-1 (KOS)		IC_50_: 0.96 and 1.92 µg/mL; SI > 218.8 and >109.4 of **25** and acyclovir, respectively, CC_50_ > 210 µg/mL.	Inhibited NF-κB activity.	
**26**	*Cis/trans*-martynoside (**26**)	*Lindernia crustacea*	EBV	P3HR-1	**-**	Inhibitory effect on the EBV lytic cycle.	[79]
**27**	*Cis/trans-* isomartynoside (**27**)						
**28**	Resveratrol (**28**)	*Polygonum cuspidatum*	HSV-1/2 (Clinical isolates)	Vero, MRC-5		Inhibited ICP4 expression.	[105]
			HSV-1 (Oral lesion, adult brain)	SKH1 mice		Inhibited HSV-induced skin lesion formation.	[106]
			HSV-1 (Oral lesion); HSV-2 (Genital lesion)			Inhibits or reduces HSV replication.	[107]
			HSV-1 (Acyclovir-resistant); HSV-2	Vero		Inhibited NF–kB activation.	[108]
			HSV-1 (KOS, 7401H, TK-deficient, PAA-resistant, acyclovir-resistant, three clinical isolates); HSV-2 (Baylor186)	Vero, female BALB/c mice	IC_50_ = 19.8, 23.3, 23.5, 24.8, 25.5 and 21.7 µg/mL against three clinical isolates, TK-deficient and PAA-resistant HSV-1, respectively.	Exhibited the inhibitory activity at the early phase and late phase of replication of HSV-1 (KOS) and HSV-2.	[109]
			HSV-1 (17); HSV-2 (G)	Vero	-	Promoted rapid and transient release of reactive oxygen species (ROS).	[110]
			VZV (Ellen)	MRC-5	EC_50_: 4 and 19 µM for acyclovir and **28** treatments, respectively.	Inhibited VZV IE62 synthesis.	[111]
			EBV	Raji, female mice	IC_50_: 16.38 µg/mL, LD_50_: 143.75 µg/ml.	Inhibited TPA-induced Epstein–Barr early antigen activation.	[112]
				P3HR-1	EC_50_ ~ 24 µM.	Preventing the proliferation of the virus.	[113]
				Raji, Akata	**-**	Inhibited protein synthesis, decreased reactive oxygen species (ROS) levels, and suppressed the EBV-induced activation of the redox-sensitive transcription factors NF–kB and AP-1.	[114]
				B95-8, Akata, B cells	**-**	Downregulation of the anti-apoptotic proteins Mcl-1 and survivin.	[115]
			HCMV (Towne, AD169)	HEL 299	IC_50_: 1.7 μM, CC_50_ > 400 μM, SI ≥ 50.	Blocked virus-induced activation of the EGFR and phosphatidylinositol-3-kinase signal transduction.	[116]
			KSHV	HEK293, BCBL-1	**-**	Lowered ERK1/2 activity, Egr1 expression.	[117]
				BCBL-1, BC-1, P3HR1, BJAB	**-**	Inhibited HHV8 gene expression and replication	[118]
**29**	Epigallocatechin gallate (**29**)	*Camellia* *sinensis*	HSV-1 (KOS)	Vero	CC_50_: 100 μM, EC_50_: 2.5 μM, and SI: 40.	Anti-HSV-1 activity.	[69]
			HSV-1 (F1); HSV-2 (333)	Vero, CV-1	**-**	Targeted gB, gD, or another enveloped glycoprotein.	[119]
					**-**	Inactivated Class I, II, and III fusion proteins of enveloped viruses.	[120]
			HSV-1 (17)	Vero	-	p-EGCG inhibited HSV-1 production via viral adsorption in vitro.	[121]
			HSV-1/2 (Clinical isolates)	Vero, MDCK	**-**	**I**nhibited virion surface proteins.	[122]
			HSV-1 (KOS)	Vero	-	Direct virucidal properties on HSV-1.	[123]
				Vero, OC3	-	Reduced the levels of viral particles and viral DNA during viral entry phase.	[124]
			EBV	P3HR1	**-**	Inhibited the expressions of EBV lytic proteins.	[125]
				B95.8, CNE1-LMP1	**-**	Involve the suppression of the activation of MEK/ERK1/2 and PI3-K/Akt signaling.	[126]
				B95.8, CNE1-LMP1	IC_50_: 20 µM.	Involving in downregulation of LMP1.	[127]
			KSHV	BCBL-1, BC-1	**-**	Induced cell death and ROS generation.	[128]
				BC-3	**-**	**D**isplayed the activity against lytic KSHV infection.	[94]
**30**	(+)-Rutamarin (**30**)	*Ruta graveolens*	EBV	P3HR-1	**-**	Exhibited anti-EBV lytic DNA replication.	[129]
			KSHV	BCBL-1, JSC-1, BJAB	IC_50_: 1.12 μM, EC_50_: 1.62 μM, SI: 84.14 and CC_50_: 94.24 μM.	Inhibition of the catalytic activity of human topoisomerase II.	[130]
**31**	Ginkgolic acid (**31**)	*Ginkgo Biloba*	HSV-1 (MacIntyre); HSV-2 (MS)	A549	-	Anti-HSV-1 and 2 activities before viral adsorption to cell surface.	[131]
			HSV-1 (Acyclovir-resistant-GFP-17+)	Vero, BALB/cJ female mice	**-**	Virucidal activity and fusion inhibition.	[132]
			HSV-1 (F); HCMV (CH19, B16)	HEp2, 293T, HFF	**-**	Inhibited viral fusion.	[133]
**32**	Emodin (**32**)	*Polygonum cuspidatum*	HSV-1 (17)	Vero	EC5_0_ = 21.54 μM in plaque reduction assay.	Inhibited nuclease activity of HSV-1 UL12.	[134]
			HSV-1 (F); HSV-2 (333)	HEp-2, specific pathogen-free BALB/c mice	-	Inhibit the replication of HSV-1 and HSV-2.	[135]
			HSV-1 (Laboratory)	HeLa, male BALB/c mice	-	Decreased TLR3 pathway and its downstream molecules.	[136]
			EBV	P3HR-1	-	Inhibitory effect on the EBV lytic cycle.	[79]
					EC_50_: 1.2 μg/mL.	Inhibit the transcription of EBV immediate early genes, the expression of EBV lytic proteins and reduces EBV DNA replication.	[137]
				NA, HA, TW01, HONE-1	CC_50_: 31, 58, 65, 79 μM for Tw01, HONE-1, HA, and NA, respectively.	Restricting EBV reactivation and NPC recurrence.	[138]
			HCMV (AD169, ganciclovir-resistant)	MRC-5	EC_50_: 4.1 and 3.7 μM/IC_50_ = 9.6 and 12.6 for HCMV AD-169 and ganciclovir-resistant strain, respectively	Anti-HCMV activity.	[139]
**33**	Aloe-emodin (**33**)	*Lindernia* *crustacea*	EBV	P3HR-1		Inhibitory effect on the EBV lytic cycle.	[79]
			HCMV (AD169)	MRC-5	EC_50_ > 37.0 μM and IC_50_ > 37.0 μM.	Anti-HCMV activity	[139]
**34**	Daunorubicin (**34**)	*Streptomyces peucetius*	KSHV	BCBL-1, HEK293, Vero, PAN-LUC	-	Induced the luciferase expression under the control of the PAN or RTA promoters, induced the expressions of lytic genes.	[140]
**35**	Doxorubicin (**35**)						
**36**	Epirubicin (**36**)						
**37**	Allicin (**37**)	*Allium sativum*	KSHV	BC-3		Antiviral activity against latent and lytic KSHV.	[94]
**38**	Artemisinin-egonol (**38**)	*Artemisia annua*	HCMV (AD169-GFP)	Type A-positive human erythrocytes	EC_50_ = 0.17 and 0.13 μM, respectively	Strong anti-HCMV activity.	[141]
**39**	Artemisinin-homoegonol (**39**)						
**40**	Byzantionoside B (**40**)	*Lindernia* *crustacea*	EBV	P3HR-1		Inhibitory effect on the EBV lytic cycle.	[79]
**41**	(+)-Hyperjaponicol B (**41**)	*Hypericum japonicum*	EBV	B95-8	EC_50_: 0.57, 0.49, 2.86 μM, SI > 52.63, 106.78, 104.50 for 41, 42, and ganciclovir, respec-tively.	Inhibited EBV DNA replication.	[142]
**42**	Hyperjaponicol D (**42**)						
**43**	Hyperjaponicol H (43)		EBV	B95-8	EC_50_: 25.00 µM; SI > 2., CC_50_ > 50 µM	Moderately inhibited EBV lytic DNA replication.	[143]
**44**	(+)-Japonicol B (**44**)		KSHV	Human-iSLK.219	EC_50_: 8.75 μM; SI = 16.06, CC_50_ = 140.60 μM.	Moderately exhibited anti-KSHV activities.	[144]
**45**	(+)-Japonicol E (**45**)		KSHV	Human-iSLK.219, Vero	IC_50_: 8.3 and 4.9 μM, SI = 23.49 and 25.7 for **45** and **46,** respectively.	Inhibitory effects on KSHV lytic replication.	[145]
**46**	(+)-Japonicol H (**46**)						
**47**	(+)-Japonone A (**47**)		KSHV	Human-iSLK.219, Vero	IC_50_: 166.0 μM; SI > 3.01, CC_50_ > 500 μM.	Inhibitory effect on KSHV lytic replication	[146]
**48**	Japopyrones B(**48**)		KSHV	Vero	IC_50_: 29.46 µM, CC_50_ > 200, SI > 6.79.	Inhibitory effect on TPA-induced KSHV lytic replication.	[147]

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
