# Peer review of "Natural Products and Their Derivatives against Human Herpesvirus Infection"

_molecules, 2021, doi:10.3390/molecules26206290_

Round 1

Reviewer 1 Report

I have read the manuscript „Natural Product and their derivatives against human herpesvirus infection”. This manuscript is very interesting and refers to a current topic.

The comments I have:

  1. The title of the manuscript should be corrected: „Natural Products and their derivatives against human herpesvirus infection”.
  2. In the text in chapter 5 and in tables, mainly in 1 but also in table 2, exactly the same information is repeated. The "Outcome" column should be removed from the tables and the information and description should be left in the text in chapter 5. The same content should not be duplicated in the text and in the tables.
  3. In Table 1 it should be Streptomyces griseus not Streptomyce griseus.
  4. There is no reference 150 in the text.

Author Response

  1. Response to Reviewer’s comments

I have read the manuscript „Natural Product and their derivatives against human herpesvirus infection”. This manuscript is very interesting and refers to a current topic.

Author’ reply: Thank you very much for your suggestion. We appreciate the time and effort you have spent to share your insightful comments, and most of your suggestions have been incorporated in the revised manuscript. Those changes are taken by using the track changes mode in MS Word.

The comments I have:

  1. The title of the manuscript should be corrected: „Natural Products and their derivatives against human herpesvirus infection”.

Author’ reply: The title of the manuscript has been corrected according to Reviewer’s comments.  

  1. In the text in chapter 5 and in tables, mainly in 1 but also in table 2, exactly the same information is repeated. The "Outcome" column should be removed from the tables and the information and description should be left in the text in chapter 5. The same content should not be duplicated in the text and in the tables.

Author’ reply: The "Outcome" column had been removed in the revised manuscript.  Authors have reorganized the table to aid the readers with understanding the topic at a glance, and the columns of “inhibitory concentration” and “mode of action” have been added as those published reviews in the similar topics (Viruses 2021, 13, 1386; Viruses 2020, 12, 154; etc).

  1. In Table 1 it should be Streptomyces griseus not Streptomyce griseus.

Author’ reply: The error had been corrected in the revised manuscript.

  1. There is no reference 150 in the text.

Author’ reply: The error had been corrected in the revised manuscript.

Reviewer 2 Report

Although the manuscript presents a good topic and is well written and organized, the manuscript needs further improvement. Therefore, I recommend some points to be considered during the revision.

The authors did not cover all recently published natural compounds with anti-herpesvirus infections. Thus, I recommend the authors use the following recommended references and add the missed natural products-derived molecules. This will help improve the quality of the paper.

1. Phenolic compounds from Morus alba such as kuwanon C, T, U, and E with anti-HSV-1 and anti-HSV-2 activities (DOI: 10.1016/j.jep.2019.112296).

2. Terpenoid compounds from Eucalyptus globulus Labill such as tereticornate A and cypellocarpin C with anti-HSV-1 and anti-HSV-2 activities, respectively (DOI: 10.3390/v10070360).

Finally, I recommend the authors Take all the above-mentioned recommendations into account and revise the manuscript accordingly. Also, I recommend the authors double-check the whole paper for grammatical and typing errors. 

Author Response

  1. Response to Reviewer’s comments

Although the manuscript presents a good topic and is well written and organized, the manuscript needs further improvement. Therefore, I recommend some points to be considered during the revision.

Author’ reply: Thank you very much for your suggestion. We appreciate the time and effort you have spent to share your insightful comments, and most of your suggestions have been incorporated in the revised manuscript. Those changes are taken by using the track changes mode in MS Word.

The authors did not cover all recently published natural compounds with anti-herpesvirus infections. Thus, I recommend the authors use the following recommended references and add the missed natural products-derived molecules. This will help improve the quality of the paper.

  1. Phenolic compounds from Morus alba such as kuwanon C, T, U, and E with anti-HSV-1 and anti-HSV-2 activities (DOI: 10.1016/j.jep.2019.112296).
  2. Terpenoid compounds from Eucalyptus globulus Labill such as tereticornate A and cypellocarpin C with anti-HSV-1 and anti-HSV-2 activities, respectively (DOI: 10.3390/v10070360).

Author’ reply: The recommended references and the missed natural products-derived molecules have been added to the revised manuscript.

Finally, I recommend the authors Take all the above-mentioned recommendations into account and revise the manuscript accordingly. Also, I recommend the authors double-check the whole paper for grammatical and typing errors. 

Author’ reply: The manuscript was edited for correct English language, grammar, punctuation, and phrasing by a language service provider. Most of grammatical and typing errors have been revised.

Reviewer 3 Report

This manuscript summarizes anti-herpesvirus effects of components derived from natural plants. It reviews many research results and is of great value to researchers in this field. However, I would like to make some comments regarding how this manuscript could be made even better.

 In Table 1 and Fig. 1-6, the functions and effects of components from plants whose structural formulae have been clarified are summarized. On the other hand, the effects of crude extracts from plants are also summarized in Table 2, and this information has not been fully covered in past research. Therefore, I would like to suggest that the description of the crude extracts be excluded and the authors concentrate on the former description.

Readers of this manuscript are likely to want to compare the antiviral activities among components at a glance. In Table 1 and the text, the inhibitory concentrations of the components are described, but columns for the inhibitory concentration and mode of action should be added in the Table 1 to aid in understanding.

I think that if the above points are revised, the paper will be even better.

Author Response

Response to Reviewer’s comments

This manuscript summarizes anti-herpesvirus effects of components derived from natural plants. It reviews many research results and is of great value to researchers in this field. However, I would like to make some comments regarding how this manuscript could be made even better.

Author’ reply: Thank you very much for your suggestions. We appreciate the time and effort you have spent to share your insightful comments, and most of your suggestions have been incorporated in the revised manuscript. Those changes are taken by using the track changes mode in MS Word.

 In Table 1 and Fig. 1-6, the functions and effects of components from plants whose structural formulae have been clarified are summarized. On the other hand, the effects of crude extracts from plants are also summarized in Table 2, and this information has not been fully covered in past research. Therefore, I would like to suggest that the description of the crude extracts be excluded and the authors concentrate on the former description.

Author’ reply: The description of the crude extracts had been excluded in the revised manuscript.

Readers of this manuscript are likely to want to compare the antiviral activities among components at a glance. In Table 1 and the text, the inhibitory concentrations of the components are described, but columns for the inhibitory concentration and mode of action should be added in the Table 1 to aid in understanding.

Author’ reply: The columns for the inhibitory concentration and mode of action had been added to Table 1 in the revised manuscript.